# Chain-Engineering-Based De Novo Drug Design against MPXVgp169 Virulent Protein of Monkeypox Virus: A Molecular Modification Approach

**DOI:** 10.3390/bioengineering10010011

**Published:** 2022-12-21

**Authors:** Muhammad Naveed, Muhammad Aqib Shabbir, Noor-ul Ain, Khushbakht Javed, Sarmad Mahmood, Tariq Aziz, Ayaz Ali Khan, Ghulam Nabi, Muhammad Shahzad, Mousa Essa Alharbi, Metab Alharbi, Abdulrahman Alshammari

**Affiliations:** 1Department of Biotechnology, Faculty of Science & Technology, University of Central Punjab, Lahore 54590, Pakistan; 2School of Food & Biological Engineering, Jiangsu University, Zhenjiang 212013, China; 3Institute of Basic Medical Sciences, Khyber Medical University, Peshawar 25120, Pakistan; 4Department of Biotechnology, University of Malakand, Chakdara 18800, Pakistan; 5Institute of Nature Conservation, Polish Academy of Sciences, 31-120 Krakow, Poland; 6School of Biological Sciences, Health and Life Sciences Building, University of Reading, Reading RG6 6AX, UK; 7Ministry of Health, Kingdom of Saudi Arabia, Riyadh 11525, Saudi Arabia; 8Department of Pharmacology and Toxicology, College of Pharmacy, King Saud University, P.O. Box 2455, Riyadh 11451, Saudi Arabia

**Keywords:** drug discovery, drug design, MPXVgp169, molecular modification, monkeypox virus

## Abstract

The unexpected appearance of the monkeypox virus and the extensive geographic dispersal of cases have prompted researchers to concentrate on potential therapeutic approaches. In addition to its vaccine build techniques, there should be some multiple integrated antiviral active compounds because of the MPV (monkeypox virus) outbreak in 2022. This study offers a computational engineering-based de novo drug discovery mediated by random antiviral active compounds that were screened against the virulent protein MPXVgp169, as one of the key players directing the pathogenesis of the virus. The screening of these candidates was supported by the use of 72 antiviral active compounds. The top candidate with the lowest binding affinity was selected for the engineering of chains or atoms. Literature assisted to identify toxic chains or atoms that were impeding the stability and effectiveness of antiviral compounds to modify them for enhanced efficacy. With a binding affinity of −9.4 Kcal/mol after chain, the lipophilicity of 0.41, the water solubility of 2.51 as soluble, and synthetic accessibility of 6.6, chain-engineered dolutegravir was one of the best active compounds, as proved by the computational engineering analysis. This study will revolutionize the era of drug engineering as a potential therapeutic strategy for monkeypox infection.

## 1. Introduction

In the middle of the coronavirus disease 2019 (COVID-19) pandemic reaching its endemic stage, a unique global monkeypox epidemic has begun to alarm the world [1]. The MPXV virus has an enclosed double-stranded DNA genome that is around 190 kb in size. It is a member of the family Poxviridae and the genus Orthopoxvirus. Several human-infecting species of the genus Orthopoxvirus include the variola virus, MPXV, vaccinia virus, and cowpox virus [2]. The monkeypox virus (MPXV) is the causative agent of the zoonotic disease monkeypox (MPX). Although MPX is a prevalent disease in parts of the west and central Africa, its recent occurrence in several non-endemic areas outside of Africa has raised serious concerns [3].

In 2003, the United States of America reported the first monkeypox epidemic outside of Africa, which was connected to contact with infected pet prairie dogs [4]. Although in-stances may have been spreading in Europe for some months, the first verified case of the 2022 worldwide pandemic was discovered on 6 May 2022, in an adult with travel connections to Nigeria. More than 30,000 cases had been reported globally as of the first week of August 2022 [5]. Depending on the lineage of the MPV strain causing the infection and the accessibility to modern healthcare, the fatality rate ranges from 1 to 10%. Most individuals recover without treatment since the symptoms of monkeypox sickness are often minor. According to CDC (Centers for Disease Control and Prevention) recommendations, infections with the monkeypox virus do not yet have a particular therapy [6]. In vitro and preclinical investigations have shown that the antiviral drug cidofovir (Vis-tide) is effective against poxviruses by inhibiting viral DNA polymerase [7]. Tecovirimat and brincidofovir medications may be administered to very unwell monkeypox patients, although the clinical results are yet uncertain [8]. Thus, no explicit drug has been designed and approved for monkeypox specifically, which has raised the dire need to provide the necessity of monkeypox virus drugs for instantaneous and effective treatment.

Direct tests on live beings have grown considerably more challenging due to the costs involved with experimenting and current ethical regulations. In this case, in silico approaches have proved effective and have developed into potent instruments for the pursuit of illness cures [9]. Since conventional drug discovery is both expensive and time-consuming, computer-aided drug design (CADD) methodologies provide a way to increase drug development efficiency while minimizing both time and expense [10,11]. In this study, we have proposed a chain-engineering-based drug design for monkeypox virus using homology modeling, Screening of antiviral active compounds, interaction analysis, ADMET profiling, and application of Lipinski’s rule for drug safety have been performed to evaluate the best active compound for monkeypox infection. This study will be proved as a certain therapeutic approach in the treatment procedures of the monkeypox virus infection that is causing an epidemic and may lead to a pandemic in the future, similar to COVID-19. However, in vitro and in vivo experiments are still required to be considered for maximum safety and efficacy of the designed drug.

## 2. Materials and Methods

### 2.1. Identification and Preparation of Virulent Protein

The targeted protein of the monkeypox virus addressed in this study is MPXVgp169 with accession number UTG40865.1, which was retrieved from the NCBI (National Centre for Biotechnology Information). The primary structure of the protein was converted into the tertiary structure by the utilization of the trRosetta (https://yanglab.nankai.edu.cn/trRosetta/, accessed on 17 October 2022). The tertiary structure of the selected protein was visualized by the Discovery Studio Visualizer [12].

### 2.2. Prediction of Binding Pockets

It is significant to identify or predict the binding sites present in the protein for better interaction analysis. For this purpose, COACH was used (https://zhanggroup.org/COACH/, accessed on 17 October 2022), which is a meta-server method for the recognition of active sites of the protein. It works on 2 comparative approaches: TM-SITE and S-SITE. These sites identify the ligand-binding templates from the BioLiP protein function database having sequence and substructure profiling. This step revealed the possible binding pockets lying in the protein. The PDB structure of the protein was given as the input and the COACH analysis was accomplished.

### 2.3. Validation of Tertiary Structure of Virulent Protein 

The predicted tertiary structure was validated for structural quality by PROCHECK (https://saves.mbi.ucla.edu, accessed on 18 October 2022) and the Ramachandran Plot was constructed. The ERRAT server was utilized for the estimation of structural quality score of the constructed protein. Therefore, the structural quality was assessed by the presence of Rama-favored regions in the computed RC plot by PROCHECK. The predicted tertiary structure was submitted as input to the PROCHECK server.

### 2.4. Identification of Compounds 

Different antiviral drug components and phytochemicals were identified by the literature review. A total of 72 antiviral active components including synthetic, and phytochemicals were selected for the screening purpose. The 3D structures of these compounds were retrieved from PubChem (https://pubchem.ncbi.nlm.nih.gov/, accessed on 18 October 2022) and the structures were retrieved in SDF format and saved as PDB format for further utilization (Protein Databank) [13].

### 2.5. Screening of Compounds

All the retrieved antiviral active components were screened through multiple ligands docking by PyRx. This is a virtual screening software for computational drug designing and discovery, which screens out libraries of candidates against the drug targets. The compound with the highest binding affinity was selected for further optimizations and the removal of toxic molecules. An Excel file in dsv format was retrieved containing the interaction parameters and binding energy.

### 2.6. Chain Optimization of the Best-Screened Active Compounds

Chain optimization is the most recent and advanced breakthrough in drug design and discovery. The unstable and toxic chains or atoms can be substituted through new chains or atoms which have the ability to stabilize the structure of the drug candidate or to enhance the pharmacophore efficiency of a drug by removing its destabilizing or toxic elements [14]. The Swiss Bioisostere from the Swiss Drug Design tool kit (https://www.expasy.org/resources/swissdrugdesign, accessed on 18 October 2022) and it was exploited for the optimization of chains from the 4 C-terminals, 2 Carbons on benzene ring and OH-terminals of the selected component. CH_2_ and Cl chains were added to stabilize the structure and improve the efficacy of the antiviral active compounds as reported in the literature.

### 2.7. Interaction Analysis 

Autodock Vina 1 revealed the docking analysis of selected best-screened active compounds [15]. This is a collection of automated docking tools that are integrated for the prediction of the interaction of small molecules with the protein. Preparation of protein and ligand was conducted, followed by the identification of active sites and grid box setting (by default). Docking was performed between the targeted protein and the chain-optimized active compound based on the lowest binding energies. The bond lengths and the types of interactions were predicted by visualizing the docked complex on PyMol.

### 2.8. Interpretation of Docking Results

Discovery Studio Visualizer was accessed for the interpretation of the docking results. It gave the type of atoms, interactions, forces, and length of bonds present in the complex. The docked complex of protein and chain-optimized best active compound was loaded onto the PLIP server and analyzed for the presence of bond types, bond angles, and other energy parameters.

### 2.9. Exploitation of Host–Pathogen Interaction Network

The network of host–pathogen interactions between the human host and the monkeypox virus was created using the PHIST [16]. It is a Phage–Host interaction search tool that employs accurate genome matches between the viral and host genomes to determine the prokaryotic hosts of viruses. In comparison to current alignment-based tools, alignment-free tools, and CRISPR-based tools, it increases host prediction accuracy at the species level (on average by 3 percentage points) (by 14–20 percentage points) [17,18,19]. PHIST is suited for metagenomics studies because it is also two orders of magnitude faster than alignment-based methods.

### 2.10. Molecular Dynamic Simulation

Molecular dynamic simulation of the docked complexes was facilitated by the IMODs server (http://imods.chaconlab.org/, accessed on 19 October 2022) [20]. It gives information about the validation of the interaction analysis and predicts the behavior of proteins when the molecule interacts with it, so that it can be simulated in the body of the host. For this analysis, the docked complex was given as input in PDB format, and the simulations were explored for further results. 

### 2.11. Pre-Clinical Testing

SwissADME is an online platform from where the drug candidate can be tested pre-clinically (http://www.swissadme.ch/, accessed on 19 October 2022) [21]. It predicts important features such as absorption, distribution, metabolism, excretion, and toxicity of the drug candidate. [19] The PDB structure of the chain-optimized best active compound was given in the input and the drug likeliness properties of the drug were calculated.

### 2.12. Validation of Lipinski’s Rule of Five

Using the Molinspiration tool, the drug-like characteristics of the chain-optimized best active compound were examined [22]. The canonical smiles of the compounds were submitted as input. Using the Molinspiration service, the molecular characteristics and bioactivity of the drugs with high affinities were predicted. Molinspiration computes the following parameters such as logP, polar surface area, mass, range of atoms, range of O or N, range of OH, range of rotatable bonds, volume, ion channel modulator, enzymes, and nuclear receptors, as well as a range of Lipinski’s rule violations [23].

## 3. Results

### 3.1. Identification and Preparation of Virulent Protein 

The MPXVgp169 protein of monkeypox with the accession number of UTG40865.1 containing 182 aa, was retrieved from NCBI (National Centre for the Biotechnology Information) and, further, it was converted into tertiary structure by trRosetta. Model 1 with the highest TM score (template modeling score) 1 and C score (structure confidence score) was 0.9, was selected and visualized by Discovery Studio as shown in Figure 1.

### 3.2. Binding Site Identification

The PDB structure of protein MPXVgp169 was given to the COACH, meta-server for the identification of actives sites of the protein. Relative methods TM-SITE and S-SITE predicted the sites and detected the ligand binding templates from the BioLiP protein function database with sequence and substructure profiling. The TM-SITE results showed a C-score of 0.21, a cluster size of 3, and the ligands were TYR, MG, and V36 with the residues 29, 88, 89, 90 and 127, respectively. The S-SITE results have a C-score of 0.20, the cluster size is 5, and the ligands are CA, SIA, and UUU with the predicted binding site residues 42, 44, 45, 49, 50, 65, 69, 92, 92, 104, 116, 122 and 161, respectively, as depicted in Figure 2.

### 3.3. Validation of Tertiary Structure of Virulent Protein 

The tertiary structure of the virulent MPXVgp169 protein was validated by the PROCHECK server. The ERRAT sever predicted the overall quality factor 91.9075 and Ramachandran plot statistics showed that 79.3% of residues are in the most favored region, 18.3% residues in the additional allowed region, 1.2% residues in generously allowed regions and 1.2% residues in disallowed regions as shown in Figure 3.

### 3.4. Screening of Compounds

The antiviral active compounds were selected and then docked against the MPVXgp169 using PyRx to filter out the docking energies of all the 72 antiviral binding energies. The binding affinities between random 72 antiviral active compounds and protein MPXVgp169 range lay between −7.5 to −4.8 Kcal/mol, respectively, as presented in Table 1. A total of 72 antiviral active compounds were analyzed on the basis of their binding affinities with the MPXVgp169. Screening out the most suitable antiviral compound using PyRx, the lowest binding affinity with MPXVgp169 was −7.5 Kcal/mol for Dolutegravir. The lowest reported binding affinity was −4.8 Kcal/mol with the Amantadine antiviral compound. The prediction of type of bonds and their lengths in docked complex are given in Table 1.

### 3.5. Chain Optimization

To boost the antiviral active compound’s potency and to stabilize the structure, CH_2_, F, and Cl chains were added using Swiss Bioisostere from the Swiss Drug Design tool kit. Unstable or toxic chains were removed and the competence of the target compound which was Dolutegravir was enhanced through chain engineering. The selected chains are shown in red, depicting the point of addition of methyl chains on 5 C-terminals, 1 fluorine atom on the OH chain of the 3rd benzene ring, and the chlorine atom on the 2nd carbon of last benzene ring on benzene ring and the OH-terminals of the selected component as picturized in the Figure 4. As described in the literature, CH_2_ chains, fluorine, and chlorine atoms can be added to the carbonyl chains or carbon atoms of benzene rings of active compounds were added to strengthen the structure, eliminate the toxicity, and hence improve the effectiveness of the antiviral active chemical (25).

### 3.6. Interaction Analysis

Molecular docking was performed between chain-optimized active compound and the targeted virulent protein MPXVgp169 using Autodock Vina 1. The docked model which was selected on the basis of the lowest binding energy retained the binding energy of −9.4 kcal/mol, predicting the more efficient binding with the protein as compared to non-chain engineered active compound whose affinity was −7.4 Kcal/mol. The docked complex of chain-engineered dolutegravir and MPXVgp169 virulent protein is presented in Figure 5.

Interaction analysis was further interpreted through discovery studio to visualize the conventional bonds by predicting the bond length. There were four hydrogen bonds with the length of 2.71, 2.42, 3.12, and 3.15 angstroms, as usually the length of a hydrogen bond is 2.7–3.2 Angstrom. Two van der Waals or hydrophobic bonds were predicted with the length of 3.37 and 3.66 angstroms as hydrophobic interaction (van der Waal bonds) have distances a bit longer, i.e., 3.3–4.0 angstroms, as signified in Table 1. Figure 6 shows the pictorial representation of the type of conventional bonds and lengths visualized by Discovery Studio.

### 3.7. Host–Pathogen Interaction

Phage–Host Interaction search tool (PHIST) predicted the host of viruses on the basis of exact matches between viral and host genomes. It improved the accuracy of host prediction at the species level. Various interactions between humans and monkey pox were depicted such as cellular interaction of anthrax toxin, cytokine signaling in immune system, IL12 signaling mediated by STAT4, IL12-mediated signaling events, IL23-mediated signaling events, and IL27-mediated signaling events with statistics-based P values as organized in Table 2. The betweenness centrality graph showed that most of the proteins were targeted between the interaction of human and monkeypox virus with total fraction of 1 indicating the significant interaction as presented in Figure 7. 

### 3.8. Molecular Dymanics Simulation

IMODs computed different parameters for the simulation analysis of the docked complex between chain-engineered active compound and MPXVgp169 virulent protein. The eigenvalue of the complex structure was predicted as 1.784310 × 10^−5^. High co-related regions in the heat map and low RMSD value indicate better interactions of the individual residues as shown in Figure 8.

### 3.9. Pre-Clinical Testing

Swiss ADMET depicted the drug-like parameters, such as physiochemical properties, water solubility, GI absorption, topological polar surface area, skin permeation, bioavailability score, synthetic accessibility, which are given in Table 3. According to the International Standard drug-likeness rules there is only one violation predicted, i.e., no. of atoms is greater than 70.

The boiled egg model delivered an intuitive, rapid, easily reproducible yet unprecedented and robust method to analyze the brain access of small molecules and the passive gastrointestinal absorption useful for drug discovery and development. If the molecule is present in the white area of the boiled egg model, it depicts gastrointestinal absorption and if the molecule is present in the yellow area in the boiled egg model, it depicts that molecule has access to the blood–brain barrier. The boiled egg model of the chain-engineered dolutegravir showed that this drug will be absorbed in the gastrointestinal tract with efficiency, as presented in Figure 9.

### 3.10. Valuation of Lipinski’s Rule 

Molinspiration calculated the parameters of Lipinski’s rule such as logP, polar surface area, mass, range of atoms, range of O or N or NH, i.e., hydrogen bond donors and range of noN, i.e., number of hydrogen bond acceptors, and range of rotatable bonds, as shown in Table 4. The description of ADMET analysis is given in Table 5.

## 4. Discussion

The precise reason for the re-emergence of MPX has not yet been determined. There is currently no known cure or vaccine against MPX; however, as was already indicated, animal research specifies that some smallpox immunizations may be useful against MPX [2]. Due to their known side effects, particularly in immunocompromised people, replicating smallpox vaccinations should not be used to immunize against MPX. Additionally, in an individual already infected with the virus, the risk of recombination between MPXV and the vaccine strain enhances [24]. According to the study carried out by Matthew W. et al. for the infections caused by the monkeypox virus, there are no particular therapies. However, because the viruses that cause monkeypox and smallpox are genetically related, medications created to treat smallpox may also be effective against monkeypox, though no specific monkeypox drug exists yet.

As antiviral therapies, numerous antiviral drugs have demonstrated some activity against various orthopoxvirus species [25]. The drug tecovirimat may also be used to treat monkeypox in people, according to limited observational evidence. In a study by Melamed et al. [26], it was reported that the tecovirimat prevented the release of viruses from the cell and has been licensed for use to treat monkeypox on the basis of results reliant on the drug’s effectiveness in pertinent animal models [26]. However, the risk of effects on human beings remains likely; also the efficacy of the drug remains unpredictable.

A lipid compound of the antiviral drug cidofovir, used to treat cytomegalovirus retinitis in AIDS patients, is called brincidofovir. This drug has been used for the treatment of poxviruses; however, its administration is only possible intravenously and has caused side effects such as nephrotoxicity [27]. When brincidofovir was delivered to prairie dogs at the same dosages as previous animal models, brincidofovir’s pharmacokinetics (PK) investigation revealed that plasma exposure (maximum concentration [C max]) was lower, suggesting that inadequate BCV exposure may account for the reduced protective impact on survival [28]. (Considering that brincidofovir is an oral antiviral, more research in humans is imperative keeping in view the results of the findings.

In this study, the screening of the most suitable antiviral active compounds that result in potential and highly competent attachments to the MPXVgp169 were analyzed by docking analysis. Additionally, preclinical testing through ADMET profiling, using computational approaches, was performed. Many significant responses of the drugs already being administered are leading to the issues being raised due to the ineffectiveness of various drugs, as the drugs are not targeted towards monkeypox specifically. A drug designed for smallpox and other orthopoxviruses cannot bind strongly to the target, while the combination and conjugation of the target antiviral compound screened in this study, chain-engineered dolutegravir, has proved to have strong binding energies and can lead to the eradication of virulence since the drug target is a virulence-causing protein of monkeypox. The aspect of the drugs being utilized for the treatment of monkeypox that was absent was the ability to progress the elimination of monkeypox precisely, as this study accomplished.

As the spread of the monkeypox virus is growing, the virus is found to evolve side-by-side. Identification and containment of the spreading outbreak depend on pre-symptomatic suspicion, rapid notification of public health authorities, and assessment of high-risk exposures [29]. A more potent and curative-focused drug has been designed and thus proposed in this paper to overcome the predictive pandemic at an early stage. Conclusively, the studies that were presented in this research contain sufficient computational pharmacological information to allow for the propagation of a specifically targeted monkeypox drug. It would be useful if in vitro research could use this drug for indication of reliability of the proposed design. Due to the noteworthy collaboration of supporting pieces of evidence in this framework, it can close the gaps that were noted in earlier research for drugs for the treatment of monkeypox.

## 5. Conclusions

The methods for preventing, containing, and treating monkeypox must change as our knowledge of the disease does. To guarantee that patients receive the finest care, treatments for monkeypox must keep up with scientific advances and management must be founded on solid, randomized evidence. As a result, it is crucial to design and discover targeted drugs in advance, and this seems to be the most logical course of action. In this study, the experimentation of chain-engineered antiviral active compounds was performed in order to enhance the ability for improvising the efficacy of the drug against MPXVgp169-virulent protein. Dolutegravir, an antiviral active compound, was selected for chain-engineering and the toxic chains, such as chains of benzene rings, were replaced with non-toxic or inert chains, such as methyl, fluorine and chlorine chains, providing it with stronger binding to the targeted receptor and increased effectiveness as a drug. The results abide by the Lipinski rule of five. Moreover, the substantiated ADMET results are additional arguments in favor of chain-engineered dolutegravir’s legitimacy as a target medication. Additionally, molecular dynamics apparently offer further credence for the findings. To confirm the outcomes indicated by this study, more in vivo and in vitro experimental assessments are required. In conclusion, the research’s findings provide enough computational pharmacological data about antiviral drugs to enable the regulation of a monkeypox treatment that is particularly targeted.

## Figures and Tables

**Figure 1 bioengineering-10-00011-f001:**
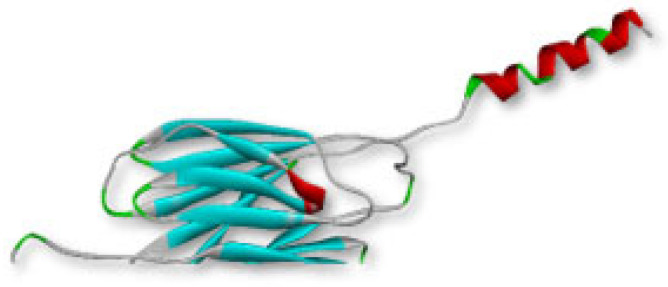
The tertiary structure of MPXVgp169 virulent protein.

**Figure 2 bioengineering-10-00011-f002:**
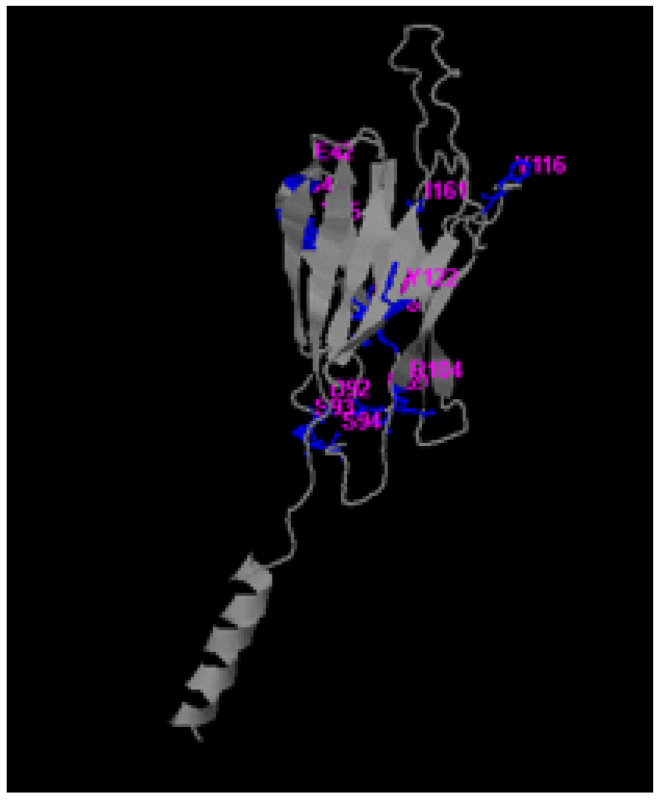
Identification of binding pockets of MPXVgp169 using TM- SITE, and S-SITE. The blue color region indicates the pocket residues, and the purple color indicates the name of the residues.

**Figure 3 bioengineering-10-00011-f003:**
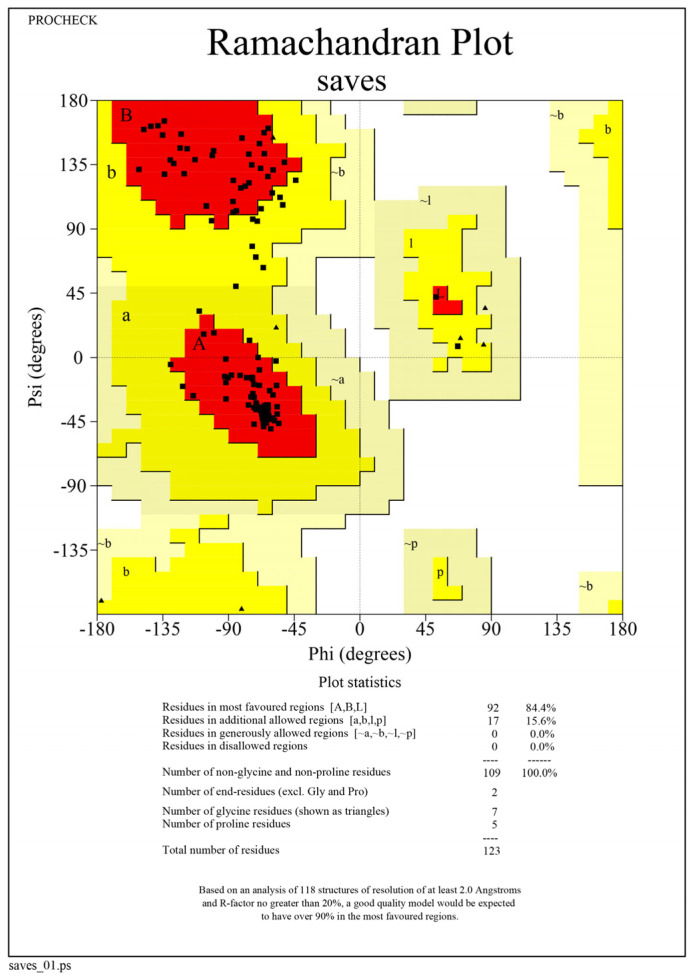
Ramachandran plot of virulent MPXVgp169 protein based on Phi and Psi angles; most favored regions (red), additional allowed regions (yellow), generously allowed regions (light brown) and disallowed regions (white).

**Figure 4 bioengineering-10-00011-f004:**
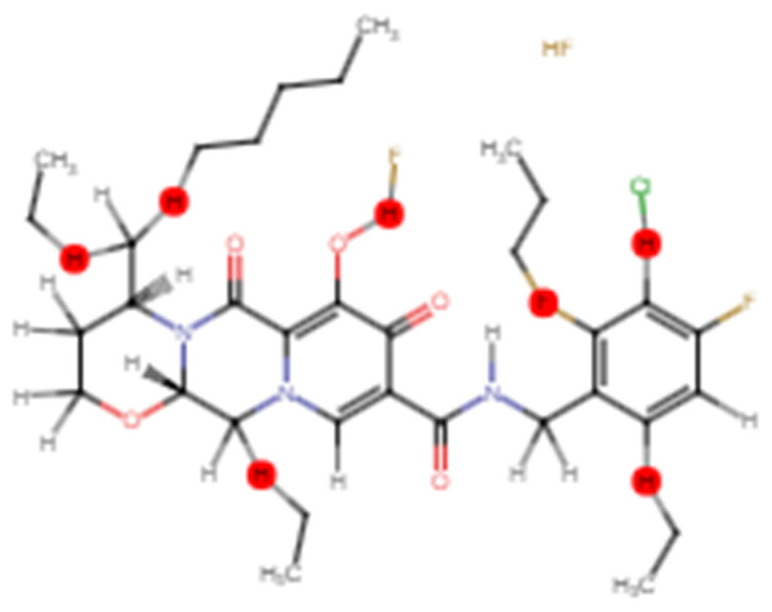
Chain engineered structure of dolutegravir; red points are representing the sites of addition of chains or atoms; brown color shows the added atom of fluorine and chlorine is presented by green color.

**Figure 5 bioengineering-10-00011-f005:**
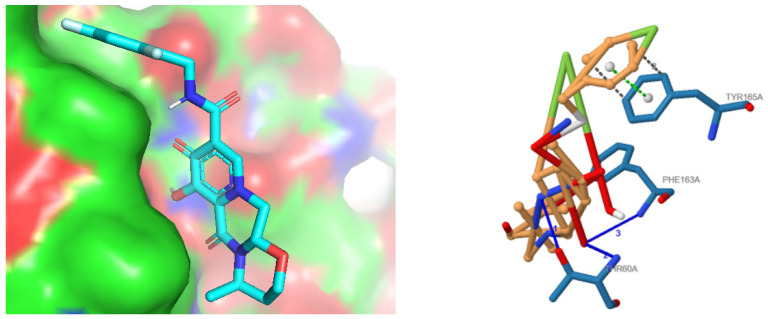
Docked complex of chain engineered dolutegravir (in ball and stick pattern) and MPXVgp169 virulent protein (in cartoon format) by Autodock Vina 1.3.7. Interpretation of Docking Analysis.

**Figure 6 bioengineering-10-00011-f006:**
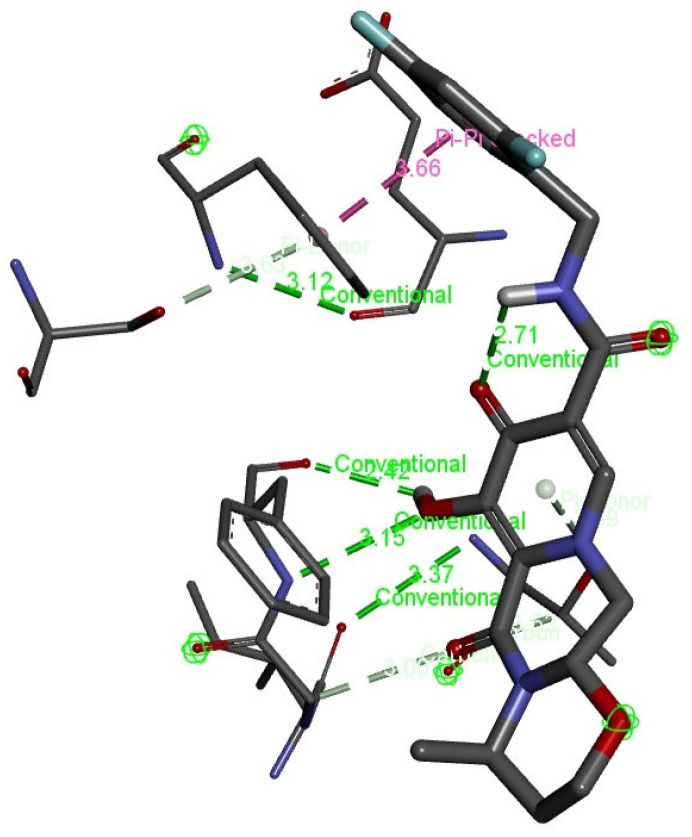
Visualization of hydrogen bonds and van der Waals interactions (dotted green lines) with bond lengths between chain-engineered active compound dolutegravir and MPXVgp169 virulent protein by Discovery Studio.

**Figure 7 bioengineering-10-00011-f007:**
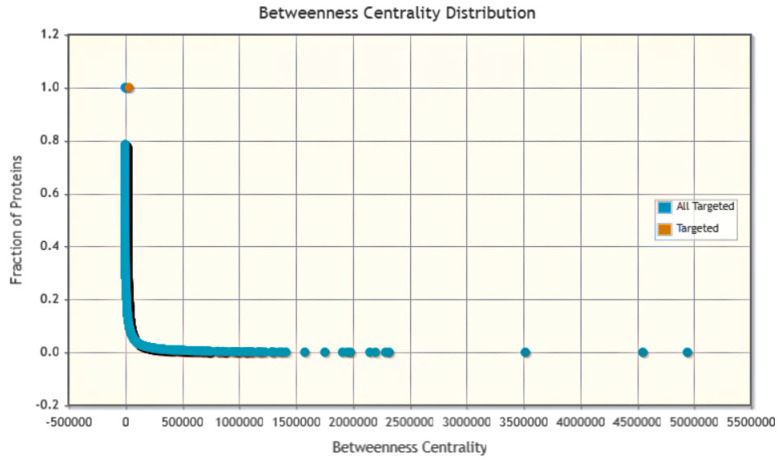
Betweenness centrality distribution graph of host pathogen interaction by PHIST.

**Figure 8 bioengineering-10-00011-f008:**
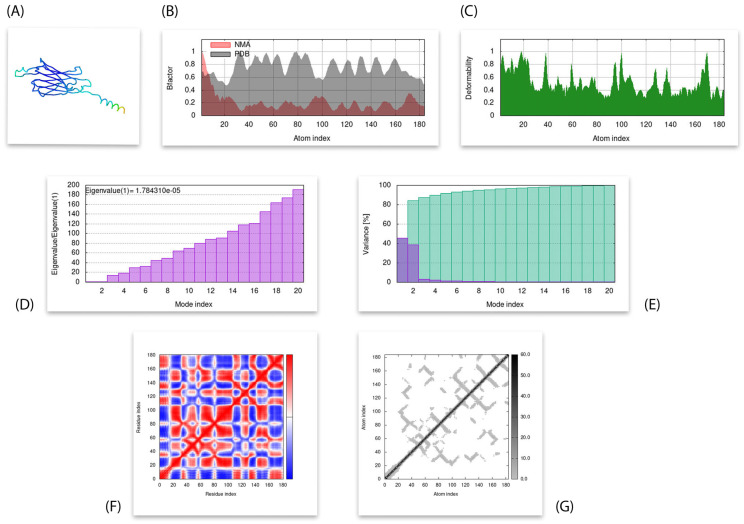
Results of molecular dynamic simulations through iMods. (**A**) Simulated 3D structure with MNA mobility; (**B**) Deformability; (**C**) B-factor; (**D**) Eigenvalues; (**E**) Variance (purple color indicates individual variances and green color indicates the cumulative variances; (**F**) Co-variance map (correlated (red) anti-correlated (blue) motions); and (**G**) Elastic network (darker grey region indicate stiffer regions).

**Figure 9 bioengineering-10-00011-f009:**
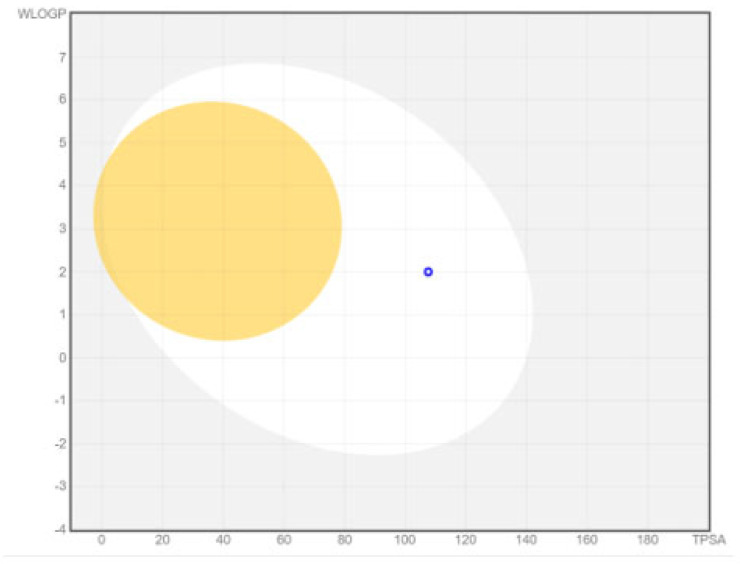
Boiled egg model of chain-engineered dolutegravir predicted by Swiss ADME; blood–brain barrier (yellow), gastrointestinal tract (white), drug (blue dot).

**Table 1 bioengineering-10-00011-t001:** Prediction of type of bonds and their lengths in docked complex through Discovery Studio.

Bond Lengths	Predicted Conventional Bonds
3.66 Angstroms	Van der Waal forces (hydrophobic interaction)
3.12 Angstroms	Hydrogen bond
2.71 Angstroms	Hydrogen Bond
2.42 Angstroms	Hydrogen Bond
3.15 Angstroms	Hydrogen Bond
3.37 Angstroms	Van der Waal (hydrophobic interaction)

**Table 2 bioengineering-10-00011-t002:** Prediction of host pathogens interaction by PHIST.

Parameters	*p* Value
Cellular roles of anthrax toxin	0.0034583
Cytokine signaling in immune system	0.037313
IL12 signaling mediated by STAT4	0.0054605
IL12-mediated signaling events	0.020386
IL23-mediated signaling events	0.012559
IL27-mediated signaling events	0.0045504

**Table 3 bioengineering-10-00011-t003:** ADMET profiling including physiochemical properties of chain-engineered dolutegravir active compound.

ADMET Parameters	Parametric Values
Formula	C20H27F2N3O5
Molecular weight	445.59 g/mol
Num. heavy atoms	30
Fraction Csp3	0.75
Num. rotatable bonds	4
Num. H-bond acceptors	7
Num. H-bond donors	2
Molar refractivity	111.68
TPSA (topological polar surface area)	107.62 Å²
Water solubility log *S* (ESOL)	−2.51
GI absorption	High
Skin permeation (Log *K*_p_)	−8.83 cm/s
Bioavailability score	0.55
Synthetic accessibility	6.65
Drug-likeness (Ghose)	1 violation i.e., #atoms > 70

**Table 4 bioengineering-10-00011-t004:** Description of parameters in the valuation of Lipinski’s rule.

Antiviral Active Compound	LogP	Molecular Weight	noN	Noh, NH
Chain-engineered dolutegravir	2.12	445.59 g/mol	5	4

**Table 5 bioengineering-10-00011-t005:** Description of parameters used in throughput methodology and their precise values.

No.	Parameter of Methods Used	Precise Values
1	Bond lengths (in interaction analysis)	Hydrogen bond (2.7–3.4 Angstroms) Van der Waal force (1–2 Angstroms)
2	Molecular weight (in ADMET analysis)	Not more than 500 g/mol
3	Num. heavy atoms	Not more than 6
4	Num. atom. low atoms	Not more than 5
5	Num. rotatable bonds	Not more than 5
6	Num. H-bond acceptors	Not more than 10
7	Num. H-bond donors	Not more than 5
8	Violation of Lipinski’s rule	Not more than 3 violations
9	LogP value	Less than 3

## Data Availability

Not applicable.

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
