# Peer review of "Chain-Engineering-Based De Novo Drug Design against MPXVgp169 Virulent Protein of Monkeypox Virus: A Molecular Modification Approach"

_bioengineering, 2022, doi:10.3390/bioengineering10010011_

Round 1
Reviewer 1 Report
The authors claim at the molecular modification approach for de novo drug design against 2 MPXVgp169 virulent protein of monkeypox virus. Although a lot of techniques are applied and interesting results are presented, the paper needs of significant efforts in order to meet the high scientific standards of such an esteemed journal as Bioengineering.
1. A lot of abbreviations, such as MPV, MPXV, CDC, aa, etc. should be specified before their use. Once introduced, abbreviations should be used instead of full explanations.
2. Row 53 –(McCarthy 2022) should be cited properly
3. Row 59 – what do the authors refers by “these medications”, since in the previous sentence only cidofovir is mentioned?
4. Section “2.4. Identification of Compounds” – “72 anti-viral active components including synthetic, and phytochemicals were selected for the screening purpose”. Are they MPXVgp169 inhibitors, as written in the abstract?
5. What is the purposing of using both PyRx and Autodock Vina for docking?
6. Section “2.8. Validation of Docking Results” sound a bit confusing. What is explained there is not a validation procedure at all…
7. What is the meaning of applying of normal mode analysis of the docked complexes?
8. Row 162 “conical” probably should become “canonical”
9. Row 173 – TM and C scores should be specified before their use
10. Figure 2 is not informative…. Rows 181-182 – “The predicted binding pockets recognized in the protein are shown in the Figure 2.”, while there is only 1 pocket visualized…
11. Errat server is not described in the methods
12. The sentence in rows 210-211 is totally incorrect… Please check and use correctly the terms for binding energies, docking scores and binding affinities…
13. I find presenting of Table 1 unnecessary… It occupies almost 2 pages, while the essential information is given in the text. So, I would suggest either presenting Table 1 in supplementary material, or even omit it.
14. Rows 222-223: “As described in the literature, CH2 chains, Fluorine and 223 Chlorine atoms can be added…” – first, there is no citation of the literature. And second, as written, it seems that might not be authors’ contribution?
15. Row 244 “doultegravir” should become “dolutegravir”
16. Figure 6 is also not informative in the presented way…
17. What is the meaning of “Interaction analysis was further validated through discovery studio to visualize the conventional bonds by predicting the bonds length.”?
18. Rows 262-263 – “Vander waal”, “Van der waal” and “Wander wall” should become “van der Waals”
19. What is the meaning of Table 2? Presented values are not assigned to any of investigated compounds?
20. The text before Figure 9 is a repetition of the figure legend.
21. There are discrepancies between the text before Table 5 and the Table itself… For clearness it should be explained what is the meaning of “noH” and “Noh, NH”…
22. Conclusions are too general…
Author Response
Dear Editor and Reviewers,
We thank you for your critical review of our manuscript and giving the chance to submit the revision. The comments were constructive, and we tried to address all of them. We attach the revised manuscript and a specific response to all the reviewers. We also highlight the changes in the manuscript. We hope that the manuscript will now be considered suitable for publication.
Reviewer 1
The authors claim at the molecular modification approach for de novo drug design against 2 MPXVgp169 virulent protein of monkeypox virus. Although a lot of techniques are applied and interesting results are presented, the paper needs of significant efforts in order to meet the high scientific standards of such an esteemed journal as Bioengineering.
Response: Thank you very much for your appreciation about our study.
- A lot of abbreviations, such as MPV, MPXV, CDC, aa, etc. should be specified before their use. Once introduced, abbreviations should be used instead of full explanations.
Response: Thank you very much for the suggestion, we have modified it. MPXVgp169 is the name of virulent protein so it doesn’t have any abbreviation.
Row 53 – (McCarthy 2022) should be cited properly
Response: Thank you very much for the correction, we have modified it in the revised manuscript.
- Row 59 – what do the authors refers by “these medications”, since in the previous sentence only cidofovir is mentioned?
Response: Thank you very much for your comment, we have added two names of the respective medications.
- Section “2.4. Identification of Compounds” – “72 anti-viral active components including synthetic, and phytochemicals were selected for the screening purpose”. Are they MPXVgp169 inhibitors, as written in the abstract?
Response: Thank you very much for the suggestion, we have modified and clarified the compounds that were used. Please see revised manuscript.
- What is the purposing of using both PyRx and Autodock Vina for docking?
Response: Thank you very much for your comment. PyRx was used for the purpose of multiple docking all at once while Autodock was used for the manual docking of best candidate dolutegravir that was extracted from Pyrx results and was chain engineered.
- Section “2.8. Validation of Docking Results” sound a bit confusing. What is explained there is not a validation procedure at all…
Response: Thank you very much for your comment. We have modified and revised it in the revised manuscript.
- What is the meaning of applying of normal mode analysis of the docked complexes?
Response: Thank you very much for pointing it out. Modification of the term “Normal Mode Analysis” has been done.
- Row 162 “conical” probably should become “canonical”
Response: Thank you very much for your correction. We have corrected it in the revised manuscript.
- Row 173 – TM and C scores should be specified before their use
Response: Thank you very much for your correction. TM and C scores have been specified in the revised manuscript.
- Figure 2 is not informative…. Rows 181-182 – “The predicted binding pockets recognized in the protein are shown in the Figure 2.”, while there is only 1 pocket visualized…
Response: Thank you very much for your suggestion. The specific pocket shown in the figure has been specified accordingly in the revised manuscript.
- Errat server is not described in the methods
Response: Thank you very much for your comment. The purpose of Errat server has been mentioned in the methods in revised manuscript.
- The sentence in rows 210-211 is totally incorrect… Please check and use correctly the terms for binding energies, docking scores and binding affinities…
Response: Thank you very much for your suggestion. The sentence has been modified accordingly in the revised manuscript.
- I find presenting of Table 1 unnecessary… It occupies almost 2 pages, while the essential information is given in the text. So, I would suggest either presenting Table 1 in supplementary material, or even omit it.
Response: Thank you very much for your suggestion. The table has been omitted from revised manuscript.
- Rows 222-223: “As described in the literature, CH2 chains, Fluorine and 223 Chlorine atoms can be added…” – first, there is no citation of the literature. And second, as written, it seems that might not be authors’ contribution?
Response: Thank you very much for the recommendation, as this idea is not solely taken from template article, this was the new concept that the authors have experimented with and got successful. Moreover, toxic domains were identified from different articles that we have mentioned in the revised manuscript now.
- Row 244 “doultegravir” should become “dolutegravir”
Response: Thank you very much for your comment. The correction has been done accordingly in the revised manuscript.
- Figure 6 is also not informative in the presented way…….
Response: Thank you very much for your comment. This is the docked complex of chain engineered dolutegravir and MXVgp169 virulent protein as mentioned in the legend of the figure.
- What is the meaning of “Interaction analysis was further validated through discovery studio to visualize the conventional bonds by predicting the bonds length.”?
Response: Thank you very much for your comment. Changes has been made accordingly.
- Rows 262-263 – “Vander waal”, “Van der waal” and “Wander wall” should become “van der Waals”
Response: Thank you very much for your correction. Correction has been made accordingly.
- What is the meaning of Table 2? Presented values are not assigned to any of investigated compounds?
Response: Thank you very much for your comment. It is the depiction of the different types of bonds between the chain engineered dolutegravir and MPXVgp169.
- The text before Figure 9 is a repetition of the figure legend.
Response: Thank you very much for your correction. Correction has been made accordingly
- There are discrepancies between the text before Table 5 and the Table itself… For clearness it should be explained what is the meaning of “noH” and “Noh, NH”…
Response: Thank you very much for your suggestion. It has been modified in the revised manuscript.
- Conclusions are too general…
Response: Thank you very much for your suggestion. Conclusion has been modified and revised in the revised manuscript. Please see revised manuscript.
With best Regards
Tariq Aziz (Postdoc, PhD)
Associate Professor
Jiangsu University China

Reviewer 2 Report
The topic is interesting, and the manuscript is reasonably organized. I thus recommend its publication after minor revision.
1. The resolution of all figures in the manuscript need to be improved.
2. English language and style should be modified.
Author Response
Dear Editor and Reviewers,
We thank you for your critical review of our manuscript and giving the chance to submit the revision. The comments were constructive, and we tried to address all of them. We attach the revised manuscript and a specific response to all the reviewers. We also highlight the changes in the manuscript. We hope that the manuscript will now be considered suitable for publication.
Reviewer 2
The topic is interesting, and the manuscript is reasonably organized. I thus recommend its publication after minor revision.
Response: Thank you very much for your appreciation about our study.
- The resolution of all figures in the manuscript need to be improved.
Response: Thank you very much for your comment. The resolution of all figures have been improved in the revised manuscript. Please see revised manuscript.
- English language and style should be
Response: Thank you very much for your suggestion. The English language, style and format has been improved. Please see revised manuscript.
With best Regards
Tariq Aziz (Postdoc, PhD)
Associate Professor
Jiangsu University China

Reviewer 3 Report
Accept after minor revision

Author Response
Dear Editor and Reviewers,
We thank you for your critical review of our manuscript and giving the chance to submit the revision. The comments were constructive, and we tried to address all of them. We attach the revised manuscript and a specific response to all the reviewers. We also highlight the changes in the manuscript. We hope that the manuscript will now be considered suitable for publication.
Reviewer 3
The authors proposed a study offers a computational engineering-based de novo drug discovery
mediated by antiviral active compounds against the virulent protein MPXVgp169, as one of the
key players directing the pathogenesis of the virus.
The objective had been reached probably, however, the results and methods needs more intensive
justification. Several notes are summarized below for better version of paper.
Response: Thank you very much for your appreciation about our study.
Major Points:
- The abstract is not clear.
Response: Thank you very much for your suggestion. The abstract has been modified in the revised manuscript. Please see revised manuscript.
- The authors have not done proper literature study. It is suggested to refer to newly
published papers and improve the literature study, with regard to metaheuristics, there is no proper literature. The authors should conduct a detailed literature study.
Response: Thank you very much for your suggestion. As this idea is not solely taken from template article, this was the new concept that the authors have experimented with and got successful. This idea has been the collaborative self-input of the authors. We have cited the article for the optimization of toxic chain.
- The authors should add section the main contributions in Motivation and contributions.
Response: Thank you very much for pointing this out. We have added a sub section after conclusion entitled Motivation in the revised manuscript. As this study is providing a solution or drug against a viral disease, which don’t have any specified medication before. So, this will motivate the researchers to take this experiment on in-vitro level for the betterment of mankind.
- There is no proper information about the datasets. There is no citations for the datasets.
Response: Thank you very much for the suggestion, we have given the URL link of PUBCHEM from where all the antiviral active compounds had been retrieved and also the citation has been added in the revised manuscript.
- The introduction section can be substantially improved with a better explanation and motivation of why the problem being solved is relevant for broader readers. Significantly, the broader readers will be interested in knowing why they should care about the proposed work. The authors are suggested to include the following research work in the literature survey for better improvement: You can use this search as a reference: An Efficient Strategy for Blood Diseases Detection Based on Grey Wolf Optimization as Feature Selection and Machine Learning Techniques
Response: Thank you very much for the suggestion, the said article has been cited in the materials and methods section. Please see revised manuscript.
- Conclusion should be improved and have concluding remarks regarding the findings of the work
Response: Thank you very much for the suggestion, It has been modified in the revised manuscript. Please see revised manuscript.
- Please add a new table that indicates a precise value of parameters of all considered methods.
Response: Thank you very much for the suggestion. We have added that table on the end of the results section please see revised manuscript.
- Pictures need improvements as fig 4,8 and 9.
Response: Thank you very much for the suggestion. Quality has been improved of the above-mentioned figures.
Minor Points:
- The footnotes of all tables should written according to the first mentioned abbreviations. It's not suggested to be written randomly since it confuses the reader.
Response: Thank you very much for the suggestion, It has been modified in the revised manuscript. Please see revised manuscript.
- In the references, try to be consistent in writing journals' names, maybe you need to look for additional resources to increase the level of evidence of your study.
Response: Thank you very much for the suggestion, For authentication we have cited the additional literature. Please see revised manuscript.
- Additionally, there are lots of grammatical mistakes and typo errors.
Response: Thank you very much for the suggestion. All mistakes, grammatical and topographical mistakes have been corrected through Grammarly.
With best Regards
Tariq Aziz (Postdoc, PhD)
Associate Professor
Jiangsu University China

Round 2
Reviewer 1 Report
The authors have done a lot of improvements it the revised version of manuscript, however, the paper still needs of some additional efforts in order to be directed to publishing.
1. Concerning my previous note
Section “2.4. Identification of Compounds” – “72 anti-viral active components including synthetic, and phytochemicals were selected for the screening purpose”. Are they MPXVgp169 inhibitors, as written in the abstract?
Authors have replied that “… we have modified and clarified the compounds that were used”, which was not actually done.
2. Concerning my previous note
Figure 2 is not informative…. Rows 181-182 – “The predicted binding pockets recognized in the protein are shown in the Figure 2.”, while there is only 1 pocket visualized…
Authors have replied that “The specific pocket shown in the figure has been specified accordingly in the revised manuscript”. In fact, Fig. 2 is the same as in the previous version.
3. Concerning my previous note
Errat server is not described in the methods
Authors have replied that “The purpose of Errat server has been mentioned in the methods in revised manuscript”, which was not actually done.
4. Concerning my previous note
The sentence in rows 210-211 is totally incorrect… Please check and use correctly the terms for binding energies, docking scores and binding affinities…
Authors have replied that “The sentence has been modified accordingly in the revised manuscript”, which was not actually done… Please now refer this note to the rows 216-217.
5. Table 1 from previous version Is omitted, but its citation is still left on row 212-213…
6. The authors do not convince me with their answer on my note concerning “As described in the literature, CH2 chains, Fluorine and 223 Chlorine atoms can be added…” – first, there is no citation of the literature. And second, as written, it seems that might not be authors’ contribution? I have seen no additions/modifications in the revised version…
7. Concerning my previous note
Figure 6 is also not informative in the presented way…
No change has been made… No active site or possible interactions or area surfaces are shown...
8. Put the citations before the dot, not after that, e.g. “. [27]” should become “[27].”
Author Response
Reviewer 1 (Round 2)
The authors have done a lot of improvements it the revised version of manuscript, however, the paper still needs of some additional efforts in order to be directed to publishing.
- Concerning my previous note
Section “2.4. Identification of Compounds” – “72 anti-viral active components including synthetic, and phytochemicals were selected for the screening purpose”. Are they MPXVgp169 inhibitors, as written in the abstract?
Authors have replied that “… we have modified and clarified the compounds that were used”, which was not actually done.
Response: In this study, 72 anti-viral compounds were screened for their MPXVgp169 inhibition activity. Molecular docking was performed to screen the compounds for their inhibition potential actually. So, the compounds with best binding energy with MPXVgp169 were considered as the potential inhibitors.
- Concerning my previous note
Figure 2 is not informative…. Rows 181-182 – “The predicted binding pockets recognized in the protein are shown in the Figure 2.”, while there is only 1 pocket visualized…
Authors have replied that “The specific pocket shown in the figure has been specified accordingly in the revised manuscript”. In fact, Fig. 2 is the same as in the previous version.
Response: The binding pockets are displayed in the Figure 3 and the figure 2 is the result of COACH server just showing the interactions. Therefore, the figure 2 has been removed and figure 3 has been renamed as figure 2 in the manuscript as well. The figure is showing all the present binding pockets in the protein.
- Concerning my previous note
Errat server is not described in the methods
Authors have replied that “The purpose of Errat server has been mentioned in the methods in revised manuscript”, which was not actually done.
Response: The purpose of ERRAT server has been added to the manuscript in heading 2.3 “Validation of Tertiary Structure of Virulent Protein” at the line 99, 100.
- Concerning my previous note
The sentence in rows 210-211 is totally incorrect… Please check and use correctly the terms for binding energies, docking scores and binding affinities…
Authors have replied that “The sentence has been modified accordingly in the revised manuscript”, which was not actually done… Please now refer this note to the rows 216-217.
Response: Apologies for the typo, we have corrected the referred lines.
- Table 1 from previous version Is omitted, but its citation is still left on row 212-213…
Response: The table 1 has been cited in the row 216.
- The authors do not convince me with their answer on my note concerning “As described in the literature, CH2 chains, Fluorine and 223 Chlorine atoms can be added…” – first, there is no citation of the literature. And second, as written, it seems that might not be authors’ contribution? I have seen no additions/modifications in the revised version…
Response: The citation has been at the end of sentence.
- Concerning my previous note
Figure 6 is also not informative in the presented way…
No change has been made… No active site or possible interactions or area surfaces are shown...
Response: The figure 6 has been replaced and modified, now the interactions with the residues are clearly visible.
- Put the citations before the dot, not after that, e.g. “. [27]” should become “[27].”
Response: Thank you for suggestion, the citation has been corrected.
Regards
Dr. Tariq Aziz (Postdoc, PhD)
Associate Professor
Jiangsu University, Zhenjiang china
